immunology, genomics

growth promotion, antibiotics, immunity trade-off, microbiome

**Author for correspondence:**
Juan A. Galarza
e-mail: juan.galarza@jyu.fi

†These authors contributed equally to the realization of this study.

# Antibiotics accelerate growth at the expense of immunity

Juan A. Galarza[1,†], Liam Murphy[1,†] and Johanna Mappes[1,2]

[1]Department of Biological and Environmental Sciences, University of Jyväskylä, Survontie, 9, P.C. 40500, Jyväskylä, Finland
[2]Organismal and Evolutionary Biology Research Program, Faculty of Biological and Environmental Sciences, Viikki Biocenter 3, FIN-00014, University of Helsinki, Finland

JAG, 0000-0003-3938-1798

Antibiotics have long been used in the raising of animals for agricultural, industrial or laboratory use. The use of subtherapeutic doses in diets of terrestrial and aquatic animals to promote growth is common and highly debated. Despite their vast application in animal husbandry, knowledge about the mechanisms behind growth promotion is minimal, particularly at the molecular level. Evidence from evolutionary research shows that immunocompetence is resource-limited, and hence expected to trade off with other resource-demanding processes, such as growth. Here, we ask if accelerated growth caused by antibiotics can be explained by genome-wide trade-offs between growth and costly immunocompetence. We explored this idea by injecting broad-spectrum antibiotics into wood tiger moth (*Arctia plantaginis*) larvae during development. We follow several life-history traits and analyse gene expression (RNA-seq) and bacterial (r16S) profiles. Moths treated with antibiotics show a substantial depletion of bacterial taxa, faster growth rate, a significant downregulation of genes involved in immunity and significant upregulation of growth-related genes. These results suggest that the presence of antibiotics may aid in up-keeping the immune system. Hence, by reducing the resource load of this costly process, bodily resources may be reallocated to other key processes such as growth.

## 1. Introduction

For the past 60 years, antibiotics have been widely used beyond the therapeutic treatment of disease, including pest control and growth promotion in a variety of taxa [1]. Worldwide, approximately 70% of all antimicrobials sold are used in animals intended for human consumption [2]. Although most research has focused on the role of antibiotics in terrestrial livestock, growth promotion through antibiotic supplementation has also been demonstrated in commercial aquaculture species [3]. In insect research, antibiotic usage has mainly focused on the treatment of bacterial infections [4], on the total or partial removal of endosymbiotic bacteria such as *Wolbachia* [5], and as potential pesticides towards some moth species [6].

In recent years, prophylactic antibiotics have been used in the raising and maintenance of large-scale insect cultures. With research laboratories aiming to maintain large numbers of insects, artificial diets and antibiotics are increasingly being used to make the rearing process more efficient. It has been realized that apart from limiting mortality due to infection, antibiotics can positively impact the growth rate of cultured insects. For instance, silkworms (*Bombyx mori*) treated with antibiotics have been found to grow to larger sizes and have heavier cocoons than untreated ones [4]. More recently, antibiotics were shown to promote the growth of several different insect species [7]. Thus, the growth-promoting effect of antibiotics seems to be common and not limited to vertebrates. Interestingly enough, there is a general lack of understanding about the molecular processes underpinning such an effect.

One way to deduce the possible mechanisms behind accelerated growth is by investigating changes in resource allocation. Because bodily resources are finite, as more resources are allocated to growth, fewer resources remain available for other processes. High resource-demanding processes are hence expected to be predominantly impacted by resource redistribution. Evidence from ecological immunity research shows that acquiring and maintaining immunocompetence is highly costly [8] and, as life-history theory suggests, expected to trade off with other important traits as a consequence [9]. This has been shown in several taxa including humans [10], insects [11], molluscs [12] and even plants [13]. Thus, we hypothesize that shifts in resource allocation between immunity and growth could potentially explain the accelerated growth observed across taxa. We investigate this hypothesis in an emerging insect model system, the wood tiger moth (*Arctia plantaginis*), through RNA-seq, 16S ribosomal profiling, as well as life-history analyses in the presence of antibiotics.

## 2. Material and methods

### (a) Study design

#### (i) Model species
The wood tiger moth (Erebidae), formerly *Parasemia plantaginis* [14], is an emerging model system with comprehensive genome resources available [15–18], and widely used in ecological and evolutionary research [19]. As adults don't feed, many factors during the larval stage, such as diet and their interaction with the environment, can have major effects later in life as adults [20,21].

#### (ii) Larval rearing
A split-family rearing design was implemented using the F1 from nine families. Each family was divided into maximum group sizes of 12 larvae, summing a total of 224 larvae each in control and antibiotic groups. The larvae were fed an artificial diet (electronic supplementary material, S1) replaced daily. Larvae in the treatment group were injected with two broad-spectrum antibiotics: tetracycline and ciprofloxacin once an instar. The larvae received three injections in total. The antibiotic solution consisted of 2 mg of tetracycline and 2 mg of ciprofloxacin dissolved into 100 ml of double distilled autoclaved water. The dose was determined by several trials using different concentrations (1 mg ml$^{-1}$, 0.5 mg ml$^{-1}$ and 0.01 mg ml$^{-1}$) following the methods of [5] until a subtherapeutic dose was obtained (0.04 mg ml$^{-1}$). The solution was injected using a sterile microneedle (10 µl) on the third last segment, parallel to the larval gut. The dose was increased by 1 µl in each subsequent injection. In parallel, larvae in the control group were pricked with an empty sterilized microneedle in the same location and for the same number of times as the treated larvae.

#### (iii) RNA-seq library construction
Twenty-four hours after antibiotics injection, one larva per family was taken for gene expression analyses and stored in RNA-later solution. A total of 24 pair-end (2 × 75 bp) cDNA libraries were constructed (four larvae/instar/treatment), according to Illumina's TruSeq mRNA-seq protocol and sequenced in an Illumina NextSeq 500 sequencer. The quality of the raw sequence reads was inspected with FastQC (https://www.bioinformatics.babraham.ac.uk/projects/fastqc/). After quality filtering and trimming, we obtained a mean of 18.2 million reads per sample, with a mean quality Phred score of 34.5 and a minimum length of 65 bp. All sequence data have been

deposited in the National Center for Biotechnology and Information (NCBI) under Bioproject PRJNA557336.

To calculate expression profiles, we aligned the high-quality filtered reads to the wood tiger moth's reference transcriptome [16]. We used the R package edgeR [22] to test for differential gene expression after each consecutive injection under a quantile-adjusted conditional maximum-likelihood (qCML) framework. Genes were considered to be significantly differentially expressed if they showed a log-fold difference of more than 2 and a *p*-value < 0.005 after a Benjamini and Hochberg correction for multiple testing [23].

We obtained a functional annotation of the expressed genes by blasting (BLASTx) [24] against a non-redundant protein database (nr) (NCBI; last accessed 06-05-2020). Gene ontology (GO) terms and information of protein family were obtained using InterProScan v. 72.0 [25]. The biological processes of the GO annotations were obtained using REVIGO [26]. A gene set enrichment analysis was performed with the R package topGO [27]. A Fisher statistic was computed using the *elim* algorithm to test for enrichment of biological processes according to the GO classification [28].

#### (iv) Gene expression validation
The RNA-seq expression profiles of a subset of four genes were validated through quantitative PCR (qPCR). We selected two genes involved in insect growth and two lipid transport genes that were up- and downregulated in all samples, respectively. As normalization controls (i.e. housekeeping genes), we selected one transcript from the RNA-seq data that showed a uniform expression level across all samples, and a gene (GADPH) known to have a stable expression in moth species in a variety of conditions [29]. The relative change in gene expression between the treatments was examined using the delta Ct method [30] taking into account multiple reference genes as described in [31].

### (b) Bacterial community analyses
To assess the effect that the antibiotic treatment had on the associated bacterial communities, we analysed the V1–V2 region of the ribosomal 16S gene sequenced in a IonTorrent PGM. Samples were taken from 42 adult males (21 per treatment) by gently squeezing their thorax with sterile tweezers to stimulate a defensive secretion from the back of their head. When attacked, adults perform a reflex bleed reaction to deter predators. Adults were chosen because they don't feed and thus taking up bacteria from the environment is unlikely at this life stage. Moreover, the secretion is an important survival trait for the species [32,33], and thus it is expected to be highly conserved, including its associated microbiota. The defensive secretion was collected under a laminar flow to minimize contamination using a sterile capillary and placed in a 1.5 ml Eppendorf tube containing 30 ul of autoclaved ddH20. DNA was extracted by inserting a metal bead (Ø 2.3 mm) inside the Eppendorf tube containing the capillary and homogenized using a bead ruptor (OMNI). After homogenization, the samples were boiled at 110°C for 10 min in a Grant heat block and stored at −20°C until further use.

Custom sequencing primers (electronic supplementary material, table S5) for amplifying the V1–V2 region (approx. 350 bp) were designed using Primer3 [34]. Sequencing libraries were prepared following the instructions of the IonTorrent 316 chip amplicon preparation protocol. A synthetic bacterial mock community was included as a sequencing control (ZymoBIO-MICS Microbial Community DNA Standard D6305) which included eight bacteria and two yeast species.

Bacterial amplicon sequence variants (ASV) were determined using the R package DADA2 following the IonTorrent pipeline [35]. Taxonomy was assigned to the ASVs according to the silva_nr99_v138.1_train_set [36]. Alpha diversity indices were obtained using phyloseq v. 1.36 [37]. We tested for differences

in the Chao1 and Shannon alpha diversity indices. The former provides estimates including estimations for unobserved taxa [38], whereas the latter considers the relative abundances of taxa and community evenness in its calculations [39]. One-way ANOVA followed by a Tukey's honestly significant difference (Tukey's HSD) *post hoc* test for pairwise comparisons were executed in R v. 4.1.0 [40]. Finally, differences in bacteria presence/absence and abundance (i.e. number of reads per taxa) between the treatments were estimated using the R package Metacoder v. 0.3.5 [41].

## (c) Life histories

We recorded the larval growth rate as the number of days from hatching until pupation, the number of days spent as pupa, the overall growth rate from larval hatching until adult eclosion, as well as the sex of the adults. Females from both treatment groups were weighed individually. Weight is commonly used as a proxy for fecundity as greater female body mass is linked to greater fecundity [42]. Additionally, we surgically extracted the eggs from the females' abdomen and counted them under a stereo microscope.

All of the analyses carried out for the life-history measurements were done in R (v. 3.5.0) [40] using the packages 'lme4' [43], 'coxme' [44] and 'MASS' [45]. Differences in growth rate were tested using a mixed-effect cox model setting the treatment group as a fixed factor and the sex as cofactor. Random factors included family and the eggs' lay date with family nested within the lay date factor as multiple families laid eggs on the same dates. Development time = treatment group + sex + (1/family/lay date).

The same formula was used when comparing time to pupation, time as a pupa and time to eclosion. Differences in mass and the mass accrued by larvae per day of development were compared using generalized mixed models following the same formula. Survival was tested by comparing survival curves by the Kaplan–Meier method and log-rank test as implemented in the R package 'survival' [46]. Differences in egg number were compared using a linear mixed-effect model which used the family as the random factor. Number of eggs = treatment group + (1 | family). As count data often violates the assumptions of linear models (i.e. linearity, normality of residuals, homoscedasticity), we performed model diagnosis analyses consisting of Shapiro–Wilk normality tests and diagnostic plots to assess the suitability of the data to our model.

# 3. Results

## (a) Gene expression profiling

A total of 93 gene transcripts belonging to growth and immunity functional categories genes were found differentially expressed (logFC > 2, $p < 0.005$) (electronic supplementary material, table S1). The full annotation including analyses, accessions, and descriptions is provided in electronic supplementary material, table S2. The most abundant annotated differentially expressed gene transcripts were chitin binding (GO:0006030) and insect cuticle proteins (GO:0042302), both related to exoskeleton (cuticle) renewal. Likewise, we identified eight gene transcripts belonging to different growth factor classes: epidermal (EGF) 5 genes, Adenosine deaminase-related (ADGF) 2 genes, Tyrosine-protein kinases 1 gene, and also growth factor antagonists such as transforming growth factors (TGF) 2 genes. Congruently, the enrichment analysis indicated a significant enrichment (Fisher $p = 0.007$) of GO terms referring to chitin metabolic processes (GO:0006030) (electronic supplementary material, table S3 and figure S1).

As the number of antibiotic injections increased, we observed a trend in the upregulation of growth-related genes (chitin, growth factors) and a downregulation of immune-related genes such as serpins, serine proteases and innate immunity genes, hereafter immune genes. After the first antibiotic injection, most of the differentially expressed genes were upregulated. This was probably as a first reaction to the injection itself. The second injection, however, started the downregulation of immune genes and the upregulation of growth-related genes (figure 1). After the third antibiotic injection, all growth-related genes except one growth factor were upregulated, and all but one immune gene were found downregulated. The same pattern of more genes being expressed late in development was also observed in absolute expression profiles (i.e. not only in differentially expressed genes; electronic supplementary material, figure S4). The exception is the first antibiotic injection, which triggered substantial gene expression, probably due to the injection itself. Hence, it is unlikely that the differences observed in gene expression could be influenced by the natural change in gene expression as development occurs.

## (b) Gene expression validation

The qPCR validation for the candidate genes was in good agreement with the RNA-sequence data showing the same pattern of up- or downregulation in both datasets (electronic supplementary material, figure S2).

## (c) Bacterial community

A total of 362 ASVs were observed distributed across samples. In the synthetic mock community, only the expected bacterial species were recovered, indicating sufficient sequencing reads and no external contamination within the sequencing run. The ANOVA analysis showed a significantly lower ($p < 0.001$) diversity in both Chao1 and Simpson alpha diversity (figure 2). The *post hoc* test indicated significant differentiation between the antibiotics and control samples for both computed indices (electronic supplementary material, table S4).

A substantial reduction in bacterial taxa was observed in the antibiotic treatment including five full phyla removed, namely Acidobacteriota, Aquificota, Cyanobacteria, Fusobacteriota and Patescibacteria. This resulted in 13 families and 41 genera absent in the antibiotic treatment (figure 3; electronic supplementary material, figure S3).

## (d) Life-histories

The larval growth rate from egg hatching to pupation differed significantly (estimate = 0.0651, $p < 0.001$), with both sexes in the antibiotic treatment reaching pupation faster than controls (figure 4). There was no significant difference in the time spent as pupa (estimate = 0.0123, $p = 0.7057$). Hence, the significant difference (estimate = 0.0589, $p < 0.001$) in growth rate from egg hatching to adult eclosion for both sexes is solely due to the faster larval growth up until pupation (figure 4; electronic supplementary material, table S7).

Control females were significantly heavier than the antibiotic-treated females (estimate = 0.0812, $p = 0.0024$) (figure 4). This could be due to the shorter development time experienced by the antibiotic larvae (figure 4), which is suggested by a low ($R = 0.32$) but significant ($p = 0.027$) correlation between development time and weight of the

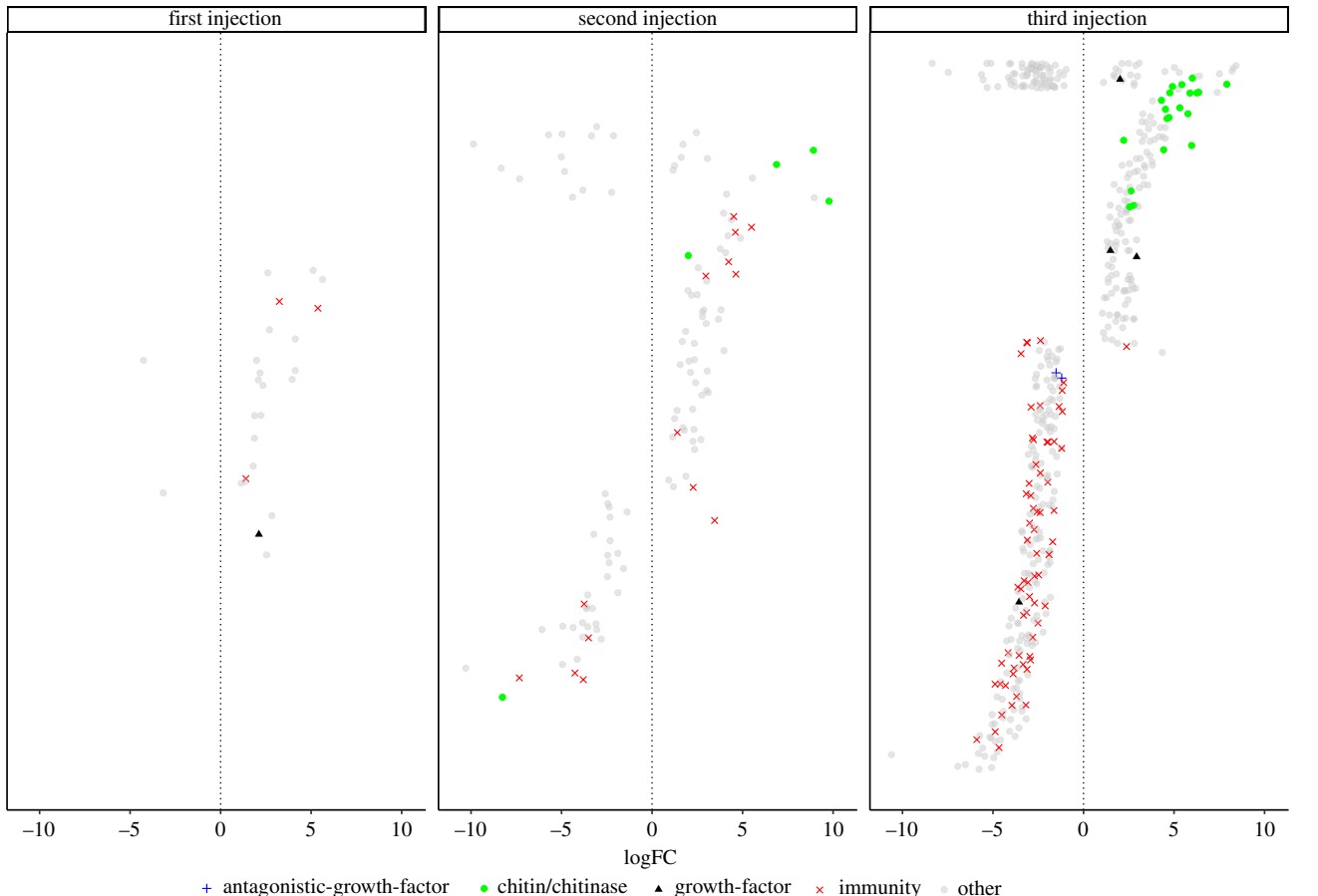

**Figure 1.** Expression patterns of growth and immune genes in the wood tiger moth (*Arctia plantaginis*) larvae after consecutive antibiotic injections. (Online version in colour.)

antibiotic-treated females (electronic supplementary material, figure S5A). The same pattern was observed between the number of eggs and female weight, which showed a strong correlation in the antibiotic treatment only (electronic supplementary material, figure S5B). However, we found no differences in the mean number of eggs (estimate 0.03308, $p = 0.2211$) after correcting for high-leverage values (figure 4; electronic supplementary material, figure S6). This suggests compensatory mechanisms for the smaller antibiotic-treated females in their reproductive output. The survival probability between the two treatments differed significantly (log-rank $p < 0.001$), particularly during the final larval instars (electronic supplementary material, figure S7). In laboratory conditions, larval mortality is typically 10–20% with the highest mortality peaks happening early and late in development. The mortality in our experiment was approximately 10% higher, most likely due to pricking itself.

## 4. Discussion

The use of antibiotics in animal husbandry has been widespread partly due to their positive effect on growth and mass gain. Nonetheless, the molecular mechanisms behind such an effect are still unclear. Here, using RNA-seq, r16S profiling and life-history analyses we investigated a potential trade-off between immunity and growth as a likely explanation. Our results suggest that the presence of antibiotics may aid in maintaining the immune system through a reduction of the bacterial load (i.e. total bacterial diversity and abundance). Hence, by reducing resources allocated for

this costly process, bodily resources may be reallocated to other key processes such as growth.

### (a) Growth

Insect growth occurs through a series of exoskeleton (cuticle) renewals or moults. Moult succession includes the separation of the cuticle from the epidermis, or apolysis, and the synthesis of a new cuticle. Chitin is a main component of the cuticle of insects providing rigidity and articulation. Its turnover is regulated by two main enzymes, chitin synthase for its synthesis and chitinase for its degradation [47]. In this study, most of chitin, chitinase and cuticle protein genes were increasingly upregulated in the antibiotic treatment, suggesting an active cuticle turnover process. This may be a response in trying to keep up with an accelerated body growth indicated by the upregulation of growth factors and the downregulation of their transforming growth factors regulators. Congruently, other studies have demonstrated that insect growth factors play important roles in larval and pupal moulting, as well as in axon ingrowth and targeting [48,49].

### (b) Immunity

The innate immune system of insects consists of physical barriers such as the integument and the peritrophic membrane, as well as humoral and cellular responses [50]. When infected, haemocytes such as plasmatocytes and granulocytes transported by the haemolymph are activated leading to phagocytosis, nodule formation and encapsulation [51]. Invading microorganisms are recognized by pattern-recognition

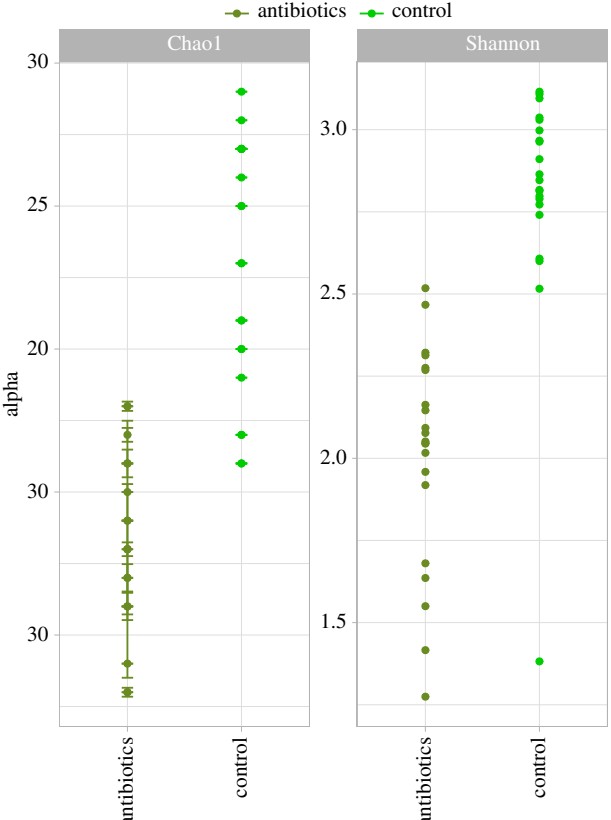

**Figure 2.** Bacterial alpha diversity indices associated with wood tiger moth (*Arctia plantaginis*) treated with antibiotics and untreated control. (Online version in colour.)

protein (PRPs) receptors that bind conserved domains located on the lipids and carbohydrates synthetized by the invading microorganisms [52]. Serine proteinases then stimulate the activation of the cytokine Spätzle and Toll pathways for the expression of antimicrobial peptides [50]. Most transcriptome studies in Lepidoptera have primarily focused on the identification and expression of immune genes as a response to bacterial and/or fungal infections [53,54]. Here, we found evidence that genes at many functional levels of the immune system (i.e. recognition, signalling and antimicrobial peptides) are responsive to antibiotics *per se*, being mainly downregulated (electronic supplementary material, table S1).

### (i) Recognition

Two PRP receptors namely C-type lectins (five genes) and scavenger receptors (three genes) were found downregulated (electronic supplementary material, table S1). Scavenger receptor genes have been previously reported upregulated in the presence of bacterial and fungal peptides in the silkworm (*Bombyx mori*) and in the diamondback moth (*Plutella xylostella*) [55,56]. In the hornworm (*Manduca sexta*), C-type lectins have been shown to bind bacterial lipopolysaccharide, inducing agglutination of bacteria and yeast, helping haemocytes eliminate infections through phagocytosis [57]. More recently, transcriptome analyses of *Gynaephora qinghaiensis* showed C-type lectins being downregulated in response to parasitism [58]. By contrast, the cabbage looper (*Trichoplusia ni*) showed a strong upregulation C-type lectins when infected by baculovirus AcMNPV [59]. Hence, it is clear that in C-type lectins induction varies according to the invader (i.e. fungi, virus or gram ± bacteria), whereas scavenger

receptors have a broader recognition spectrum. Our finding of downregulation of both C-type lectins and scavenger receptors suggests that antibiotics may help contain general infections, and thus PRPs are de-activated by the innate immune system.

### (ii) Signalling

Insects respond to infections via the Spätzle and Toll pathways, which are activated by serine proteases signalling cascades for melanization and antimicrobial peptides [60]. Serine proteases circulate as inactive zymogens in the haemolymph and become sequentially activated upon recognition of microbial polysaccharides by PRPs. Serines are inactivated by serpines after the accomplishment of their defensive functions. The balance between the effectors serines and their modulators serpins ultimately determines the susceptibility or resistance to infection [61]. In this study, serines included in the signal modulation group, like serine protease, serine proteinase, and trypsin-like serine proteinase, were found up- and downregulated. This is in agreement with other lepidopteran studies that have found differential regulation in their expression and in their serpin modulators in response to fungi [56], bacteria [62] or parasites [63]. Here, however, serines were mostly downregulated as the number of antibiotic injections increased, whereas serpins were found persistently downregulated (electronic supplementary material, table S1). This suggests that antibiotics may disrupt the immunologic balance by effectively suppressing serine regulators. This could have important consequences as an immunologic unbalance could cause complete immunosuppression, or an over-response with the consequence of self-tissue damage and/or the elimination of beneficial or commensal microbes.

### (iii) Antimicrobial peptides

In Lepidoptera, the most commonly reported peptides against various microbial infections are attacins, cecropins, lebocins, gloverins, gallerimycins, hemolyn and defensins [58,64]. In this study, no known antimicrobial peptides could be detected to have been induced. This is to be expected given the downregulation of recognition and signalling pathways that trigger their synthesis.

Previous studies have tested different immune reactions of the wood tiger moth when challenged with different microbes. Infected larvae of high and low pathogen resistance (i.e. based on cuticular melanin content), with high- and low-virulence strains of *Serratia marcescens*, were reared on diets with and without antimicrobial compounds [65]. The antimicrobial diet enhanced survival only of the high-melanin larvae, which were also more resistant to the low-virulence strain but not the high-virulence strain. In a later study, Mikonranta *et al.* [66] tested the effect of immune priming by feeding pathogenic (*S. marcescens*) or non-pathogenic (*E. coli*) bacteria to wood tiger moth larvae and injected the same bacteria 5 days later. The authors then tested for phenol bactericidal reactive oxygen species (ROS), phenoloxidase (PO) and lytic activity from the haemolymph. Larvae exposed to *S. marcescens* had higher ROS. However, lytic and PO did not differ from the *E. coli* priming. By contrast, [67] reported a high PO activity in larvae that have been fed an antimycotic (fumagillin). [68] simulated ectoparasitism by implanting nylon threads on larvae and measured the encapsulation response (i.e. darkening of the implant), as well as the lytic activity in

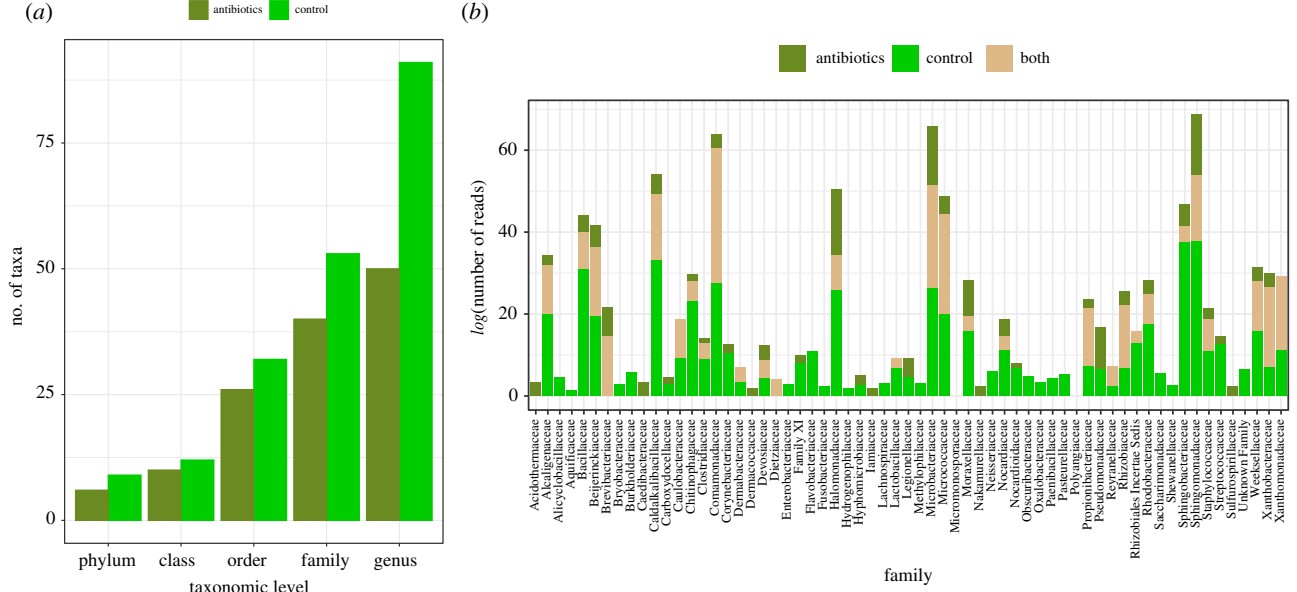

**Figure 3.** (a) Number of bacterial taxa per taxonomic level not found in wood tiger moths (*Arctia plantaginis*) treated with antibiotics. (b) Bacterial families and their abundance present in samples treated with antibiotics, or in the control group, or in both. (Online version in colour.)

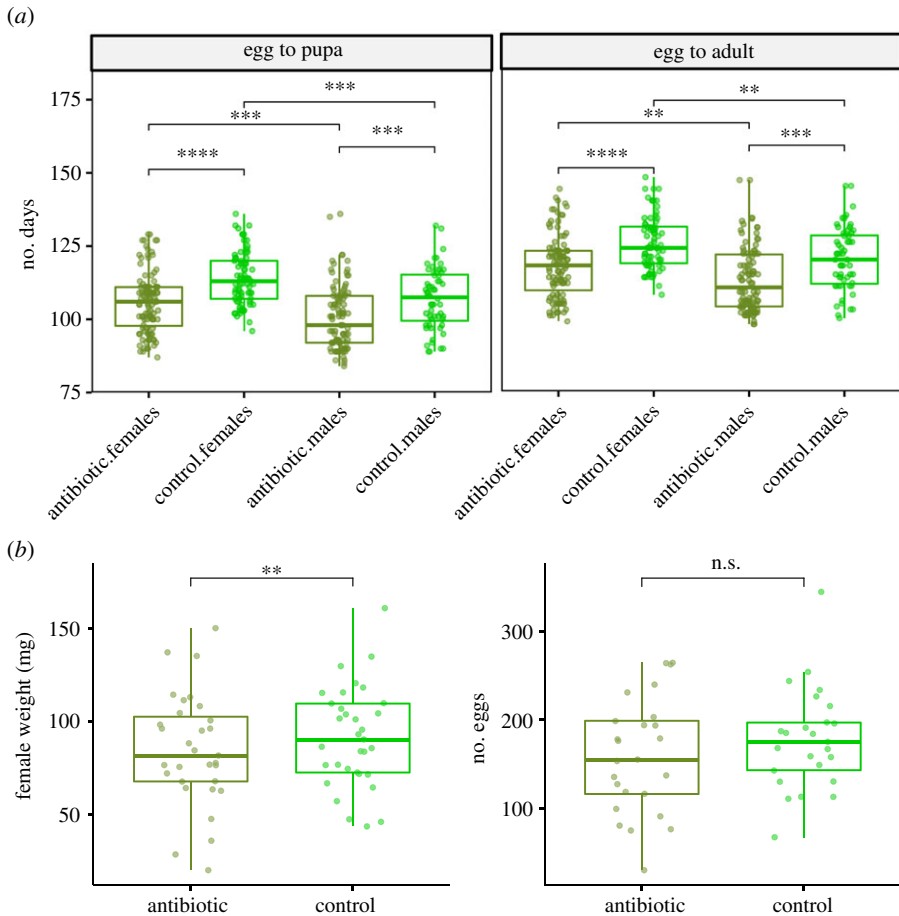

**Figure 4.** (a) Life-history traits of the wood tiger moth (*Arctia plantaginis*) under antibiotics and control treatments. The number of days elapsed from egg hatching until pupation and adult stages are shown for both sexes. Boxplot shows the median and the interquartile range to the 25th and 75th percentile. ****: $p < = 0.0001$, ***: $p < = 0.001$, **: $p < = 0.01$. (b) Adult female weight and number of eggs produced in the antibiotics and control treatments. (Online version in colour.)

the haemolymph. The results showed that adults of different colours reacted differently in their encapsulation response and lytic activity. Altogether, these previous studies indicate that wood tiger moth can mount immune reactions against

different pathogens (i.e. parasites, gram ± bacteria or fungi), and the effectiveness of such immune reactions greatly depends on host condition, colour morph, diet and pathogen virulence. In the present study, we add to this existing

knowledge by showing that the genes involved in recognizing, signalling and mounting immune responses can be suppressed in the presence of antibiotics.

## (c) Bacterial load

Microorganisms are ubiquitous in insects. It is estimated that roughly 70% of species host one or more microorganismal symbionts [69]. Microorganisms can profoundly impact insects' physiology, ecology and evolution [70]. Bacterial lineages, in particular, have evolved diverse mechanisms to gain entry and proliferate in the tissues and cells of insect hosts [71]. Bacteria can have a substantial influence on growth and immune processes. For instance, gut bacteria have a major role in providing essential nutrients for their insect host [72], whereas the hosts' immune system promotes the growth of beneficial bacteria and helps maintain a stable microbial community [73]. Hence, by perturbing the bacterial load, the cross-talk between immunity and growth can be impacted. Here, antibiotics significantly reduced bacterial abundance and diversity (figures 2 and 3). Some of the bacteria removed or depleted are known to be pathogenic for insects (i.e. entomopathogenic). Notably, bacteria from *Bacillus* and *Acinetobacter* genera are highly toxic to some Lepidoptera species [74–76] (electronic supplementary material, table S6). It can be envisaged that the depletion of pathogenic bacteria by the antibiotics freed resources allocated to keep the bacteria at bay, which could then be reallocated to growth. This is supported by the downregulation of immune genes and the upregulation of growth-related genes as a response to antibiotics. However, it is unclear if the moth's relaxed immune activity and growth increase are due to a depletion of toxigenic bacteria, or simply to a reduction in the bacterial load. An increased bacterial diversity may require higher immune responses to constrict abundances of multiple taxa that could upset homeostasis if left to multiply unchecked. These taxa would not have to be pathogenic, it could merely upset chemical balances in the microbiome and subsequent potential functions it provides to its host. Further studies using targeted antibiotics are needed to discriminate the host's immune reaction to toxins and to symbiotic bacteria. In any event, our results indicate that a disrupted microbiota can have a significant impact in the interaction between growth and immunity.

## (d) Life histories

We observed faster growth and higher survival rates of antibiotic-treated larvae, which is in agreement with previous findings in several taxa [3,77–79]. This can be advantageous for today's insect mass-rearing for different purposes like pest control, commerce and research. Aside from the advantages for the husbandry side, plasticity in growth rate should provide some advantage to the moth itself, otherwise why invest in faster growth when the chance arises? In non-seasonal environments, for instance, a faster growth rate can be seen as a major advantage for insects as the potential to die before reproduction is reduced [80].

While the control larvae took longer to develop, once they reached adulthood, the emerging females were significantly heavier than the antibiotic-treated group (figure 4b). This is congruent with previous findings where antibiotics were fed to pre-diapausing larvae of the wood tiger moth [67]. It is commonly accepted that heavier weight translates into

greater fecundity [81], as was shown by Dickel *et al.* [67], where heavier control females laid more eggs than females that had been feed with antibiotics. In the present study, however, we found no differences in the number of eggs even though control females showed a heavier weight. This suggests compensatory mechanisms presumably operating during the pupal stage in which the effect of antibiotics was not observed. Alternatively, the contrasting results could be due to different sampling strategies. In [67], the number of laid eggs was counted, whereas in this study, we counted the number of eggs inside the females. In addition, here, we use a combination of two broad-spectrum antibiotics injected into the larvae, whereas in [67], a fungicide was fed to the larvae. At the moment, it is unclear if wood tiger moth females lay all the eggs they produce. Future studies should consider the number of hatched larvae as a metric to evaluate the effect of antibiotics in both, male and female fecundity.

## 5. Conclusion

We found evidence that by perturbing the microbial community with antibiotics, resource allocation trade-offs can be generated between high-resource-demanding processes such as growth and immunity. Our main finding of downregulation of the immune system could have important implications for several taxa. For instance, while there are marked differences between insect and mammal immune systems, there are also many conserved similarities in their innate immunity due to a common evolutionary origin [82]. Both systems consist of humoral and cellular responses involving processes such as recognition, signalling cascades and antimicrobial peptide secretion. Several insect taxa (i.e. *Drosophila melanogaster, Galleria mellonella, Manduca sexta* and *Bombyx mori*) are increasingly being used in the medical field to overcome the disadvantages associated with testing in mammalian systems (e.g. cost, housing and legal/ethical restrictions) while generating comparable results [83]. Prophylactic antibiotic treatment is a common practice in the medical field. However, the interacting effects of antibiotics with other fundamental processes, such as modulation of the immune system, are surprisingly understudied at the molecular level. Thus, the paradigms set in insects can serve to guide disease development (i.e. transmission and virulence) of medically important pathogens. This is also true for commercial livestock, for which there is growing evidence of antibiotic resistance and its transmission to humans due to antibiotic administration [2]. Finally, for insect farming, either for research or conservation-management purposes, the results obtained here can inform producers about the potential negative side effects of antibiotics as growth promoters, such as immune suppression.

Data accessibility. All sequence data have been deposited in the National Center for Biotechnology and Information (NCBI) under Bioproject PRJNA557336. The code to generate statistics and figures is given in the electronic supplementary material [84].

Authors' contributions. J.A.G.: conceptualization, formal analysis, funding acquisition, investigation, methodology, project administration, supervision, visualization, writing-original draft, writing-review and editing; L.M.: data curation, formal analysis, investigation, methodology, writing-review and editing; J.M.: conceptualization, funding acquisition, methodology, project administration, resources, supervision, writing-review and editing. All authors gave final approval for publication and agreed to be held accountable for the work performed therein.

Competing interests. The authors declare no competing interests.

**Funding.** This project was funded by the Academy of Finland grant nos. 322536 to J.A.G. and 320438 to J.M.

**Acknowledgements.** The authors thank Sari Viinikainen, Kaisa Suisto, and Jimi Kirvesoja for their valuable help in insect rearing, Profs. Mikko Hurme and Joachim Kurtz for their constructive and critical comments on our work, the journal club of the Ecology and Evolutionary Biology Unit, and two anonymous reviewers for their helpful comments.

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
