## [Peer Review File · Proceedings of the Royal Society B: Biological Sciences]

Review History

RSPB-2021-0111.R0 (Original submission)

Review form: Reviewer 1

Recommendation

Reject – article is scientifically unsound

Scientific importance: Is the manuscript an original and important contribution to its field?

Good

General interest: Is the paper of sufficient general interest?

Good

Quality of the paper: Is the overall quality of the paper suitable?

Marginal

Is the length of the paper justified?

Yes

Should the paper be seen by a specialist statistical reviewer?

No

Do you have any concerns about statistical analyses in this paper? If so, please specify them explicitly in your report.

No

It is a condition of publication that authors make their supporting data, code and materials available - either as supplementary material or hosted in an external repository. Please rate, if applicable, the supporting data on the following criteria.

Is it accessible?

Yes

Is it clear?

Yes

Is it adequate?

No

Do you have any ethical concerns with this paper?

No

Comments to the Author

The aim of this paper is to test whether antibiotic treatment can alter the trade-off between growth and immunity in wood tiger moths. This is interesting as it bears on the use of antibiotics as growth promoters in animal husbandry. Using a split brood design, the authors fed half of the larvae in a set of broods with antibiotics and then measured several life-history traits as well as transcriptional profiles. They found that antibiotic fed larvae pupated and reached adulthood a few days earlier than untreated larvae; however, egg number was unchanged and treated adult females were lighter. They also found that several genes associated with immunity were down-regulated in treated larvae while genes associated with growth were upregulated.

The results are sound, but I'm not convinced they really increase understanding of why antibiotics lead to growth promotion. It also isn't known if the link between growth and immunity is causal. First, the issue of growth promotion is undoubtedly tied to the microbes in the moth. Antibiotics can reduce bacterial numbers or change microbiome composition, either of which could affect immunity. Any effects of the antibiotic treatment on bacteria were unfortunately not measured. Second, growth promotion is strongly dependent on antibiotic concentrations, how much is added and for how long the dose remains high enough to be effective. As these weren't manipulated, it isn't clear if the dose was sub-therapeutic, therapeutic or even toxic. Without estimates on the bacteria that tetracycline and ciprofloxacin target, this isn't known. I don't like asking for additional experiments, but in this case, I strongly feel that the story is very incomplete without data on the bacteria.

A few more detailed questions are given below:

- 1) A key issue is the assumption that changes to a particular set of genes are associated with a change in a phenotype of interest. How strong is the assumption that changes to a collection of immunity genes has actually had an impact on immunity?
- 2) Related to the above, gene expression data are relative, i.e. treated to untreated. Increased relative expression can therefore be caused by an absolute increase in the treatment group or an absolute decrease in the untreated group (and vice versa). As immunity genes, and obviously growth related genes, change naturally during development, it would be valuable to comment on/show absolute as well as relative differences through time.
- 3) Adult females in the antibiotic group are actually lighter, in contrast to the expectation that these agents are growth promoters. Is it possible that they are lighter simply because their developmental time is shorter, whereas the untreated slower larvae have a few extra days to feed? Given this, it might be worth checking if time to pupation and female weight are correlated, and if this scaling varies between the treatment groups (especially given the fairly sizable variance in both variables). The same question applies to the scaling between mass and egg number.
- 4) Mortality data are entirely overlapping until the last instar. Is there any reason why 35%

of the larvae in the untreated group suddenly die? Greater survival could be driven by the direct effects of the antibiotics in the antibiotic-treated larvae. But it's striking that this increased mortality occurs in the group with the higher expression of immune genes. This could be further explained.

Review form: Reviewer 2

Recommendation

Accept with minor revision (please list in comments)

Scientific importance: Is the manuscript an original and important contribution to its field?

Excellent

General interest: Is the paper of sufficient general interest?

Excellent

Quality of the paper: Is the overall quality of the paper suitable?

Good

Is the length of the paper justified?

Yes

Should the paper be seen by a specialist statistical reviewer?

No

Do you have any concerns about statistical analyses in this paper? If so, please specify them explicitly in your report.

Yes

It is a condition of publication that authors make their supporting data, code and materials available - either as supplementary material or hosted in an external repository. Please rate, if applicable, the supporting data on the following criteria.

Is it accessible?

Yes

Is it clear?

Yes

Is it adequate?

No

Do you have any ethical concerns with this paper?

No

Comments to the Author

This paper investigates whether antibiotic-induced growth promotion results from a trade-off between immune investment and growth. The authors use the wood tiger moth (*Arctia plantaginis*) to test this hypothesis. Larvae treated with broad-spectrum antibiotics showed a faster growth rate, up-regulation of growth genes, and down-regulation of genes involved in immunity. Overall, this is an interesting paper that investigates a question important to agriculture as well as to life history evolution.

Major comments:

1. The authors observe that antibiotic treatment leads to down-regulation of immune genes and increased growth. It seems to me that there are at least two possible explanations for this, which are not mutually exclusive: 1) decreased caloric investment in immunity allows for increased investment in growth (life history trade-off), or 2) decreased nutrient uptake by bacteria frees up energy for increased growth (reduced pathogen burden). The authors focus on the first possibility, but it would be worthwhile to add discussion of alternative explanations.

Minor comments:

1. For Figures 1&2, please add a brief description of what the box plots show in the legend (e.g. median and quartiles shown for each group).
2. Figure 4: Please increase the font size for the labels in this figure.
3. It would be helpful to briefly discuss whether the observed changes in growth and gene expression are mediated by the removal of bacteria or direct physiological effects of antibiotics on invertebrate cells. For example, is growth promotion observed when insects are raised in germ-free conditions? This would suggest that decreased bacterial load is important to the mechanism of growth promotion. In vertebrates, antibiotics like macrolides can also have direct physiological effects on cells in the absence of bacteria.
4. Data availability is listed for sequencing data but not life-history data (e.g. growth data and egg counts). Please make these data available.
5. Linear mixed effects models are used here to analyze count data (e.g. egg numbers). Count data often violate the assumptions of a linear model, and a Poisson model might be more appropriate. Have model diagnostics shown that the linear model is appropriate to these data? If not, I would suggest a Poisson distribution or similar alternative.

Decision letter (RSPB-2021-0111.R0)

16-Feb-2021

Dear Dr Galarza:

I am writing to inform you that your manuscript RSPB-2021-0111 entitled "Antibiotics accelerate growth at the expense of immunity" has, in its current form, been rejected for publication in Proceedings B.

This action has been taken on the advice of referees, who have recommended that substantial revisions are necessary. With this in mind we would be happy to consider a resubmission, provided the comments of the referees are fully addressed. However please note that this is not a provisional acceptance.

Sincerely,
 Professor Hans Heesterbeek
 mailto: proceedingsb@royalsociety.org

Associate Editor

Comments to Author:

Thank you for submitting your manuscript "Antibiotics accelerate growth at the expense of immunity" to Proceedings B. I have now received two reviews and evaluated the manuscript myself. We all see clear importance in the question investigated, however, less clear is the level of evidence that this study provides and its ability to distinguish between multiple explanations; see in particular comments from reviewer 1 regarding the effects of antibiotics on bacterial infections. In addition to points raised by the reviewers I'm wondering why only needle prick was used for the control group, rather than injection of distilled water, and if this could have influenced the results.

Reviewer(s)' Comments to Author:

Referee: 1

Comments to the Author(s)

The aim of this paper is to test whether antibiotic treatment can alter the trade-off between growth and immunity in wood tiger moths. This is interesting as it bears on the use of antibiotics as growth promoters in animal husbandry. Using a split brood design, the authors fed half of the larvae in a set of broods with antibiotics and then measured several life-history traits as well as transcriptional profiles. They found that antibiotic fed larvae pupated and reached adulthood a few days earlier than untreated larvae; however, egg number was unchanged and treated adult females were lighter. They also found that several genes associated with immunity were down-regulated in treated larvae while genes associated with growth were upregulated.

The results are sound, but I'm not convinced they really increase understanding of why antibiotics lead to growth promotion. It also isn't known if the link between growth and immunity is causal. First, the issue of growth promotion is undoubtedly tied to the microbes in the moth. Antibiotics can reduce bacterial numbers or change microbiome composition, either of which could affect immunity. Any effects of the antibiotic treatment on bacteria were unfortunately not measured. Second, growth promotion is strongly dependent on antibiotic concentrations, how much is added and for how long the dose remains high enough to be effective. As these weren't manipulated, it isn't clear if the dose was sub-therapeutic, therapeutic or even toxic. Without estimates on the bacteria that tetracycline and ciprofloxacin target, this

isn't known. I don't like asking for additional experiments, but in this case, I strongly feel that the story is very incomplete without data on the bacteria.

A few more detailed questions are given below:

- 1) A key issue is the assumption that changes to a particular set of genes are associated with a change in a phenotype of interest. How strong is the assumption that changes to a collection of immunity genes has actually had an impact on immunity?
- 2) Related to the above, gene expression data are relative, i.e. treated to untreated. Increased relative expression can therefore be caused by an absolute increase in the treatment group or an absolute decrease in the untreated group (and vice versa). As immunity genes, and obviously growth related genes, change naturally during development, it would be valuable to comment on/show absolute as well as relative differences through time.
- 3) Adult females in the antibiotic group are actually lighter, in contrast to the expectation that these agents are growth promoters. Is it possible that they are lighter simply because their developmental time is shorter, whereas the untreated slower larvae have a few extra days to feed? Given this, it might be worth checking if time to pupation and female weight are correlated, and if this scaling varies between the treatment groups (especially given the fairly sizable variance in both variables). The same question applies to the scaling between mass and egg number.
- 4) Mortality data are entirely overlapping until the last instar. Is there any reason why 35% of the larvae in the untreated group suddenly die? Greater survival could be driven by the direct effects of the antibiotics in the antibiotic-treated larvae. But it's striking that this increased mortality occurs in the group with the higher expression of immune genes. This could be further explained.

Referee: 2

Comments to the Author(s)

This paper investigates whether antibiotic-induced growth promotion results from a trade-off between immune investment and growth. The authors use the wood tiger moth (*Arctia plantaginis*) to test this hypothesis. Larvae treated with broad-spectrum antibiotics showed a faster growth rate, up-regulation of growth genes, and down-regulation of genes involved in immunity. Overall, this is an interesting paper that investigates a question important to agriculture as well as to life history evolution.

Major comments:

1. The authors observe that antibiotic treatment leads to down-regulation of immune genes and increased growth. It seems to me that there are at least two possible explanations for this, which are not mutually exclusive: 1) decreased caloric investment in immunity allows for increased investment in growth (life history trade-off), or 2) decreased nutrient uptake by bacteria frees up energy for increased growth (reduced pathogen burden). The authors focus on the first possibility, but it would be worthwhile to add discussion of alternative explanations.

Minor comments:

1. For Figures 1&2, please add a brief description of what the box plots show in the legend (e.g. median and quartiles shown for each group).
2. Figure 4: Please increase the font size for the labels in this figure.
3. It would be helpful to briefly discuss whether the observed changes in growth and gene expression are mediated by the removal of bacteria or direct physiological effects of antibiotics on invertebrate cells. For example, is growth promotion observed when insects are raised in germ-free conditions? This would suggest that decreased bacterial load is important to the mechanism of growth promotion. In vertebrates, antibiotics like macrolides can also have direct physiological effects on cells in the absence of bacteria.

4. Data availability is listed for sequencing data but not life-history data (e.g. growth data and egg counts). Please make these data available.

5. Linear mixed effects models are used here to analyze count data (e.g. egg numbers). Count data often violate the assumptions of a linear model, and a Poisson model might be more appropriate. Have model diagnostics shown that the linear model is appropriate to these data? If not, I would suggest a Poisson distribution or similar alternative.

Author's Response to Decision Letter for (RSPB-2021-0111.R0)

See Appendix A.

RSPB-2021-1819.R0

Review form: Reviewer 1

Recommendation

Accept with minor revision (please list in comments)

Scientific importance: Is the manuscript an original and important contribution to its field?

Good

General interest: Is the paper of sufficient general interest?

Good

Quality of the paper: Is the overall quality of the paper suitable?

Good

Is the length of the paper justified?

Yes

Should the paper be seen by a specialist statistical reviewer?

No

Do you have any concerns about statistical analyses in this paper? If so, please specify them explicitly in your report.

No

It is a condition of publication that authors make their supporting data, code and materials available - either as supplementary material or hosted in an external repository. Please rate, if applicable, the supporting data on the following criteria.

Is it accessible?

Yes

Is it clear?

Yes

Is it adequate?

Yes

Do you have any ethical concerns with this paper?

No

Comments to the Author

The new results in this resubmission, especially data on changes to the microbiome during antibiotic treatment, significantly improve the manuscript. They also lead to a few new questions and suggestions.

- 1) Please clarify that defensive secretions contain microbes that are representative of the gut population. I assume this is the case, but I don't know the system well enough to be sure.
- 2) Antibiotics can cause changes to total bacterial densities, microbiome composition or both. Although the data focus on microbial diversity, the response letter and Discussion (e.g. Line 348) seem to focus on "bacterial lode" (which I assume means total density/abundance).
 - a. Abundance isn't measured, as far as I can tell. This should be clarified.
 - b. Several diversity indices are shown in Fig 2. As these are all concordant, it isn't really necessary to show them all. More importantly, please spell out the relevance of diversity per se (even if this is only a guess). Is the argument that reduced diversity results in the loss of bacterial groups with possible pathogens, or that higher diversity requires a more "expensive" immune response?
 - c. Fig 3 is not very informative as presented. Both somewhat duplicate the information in Fig 2 by showing that diversity is lower across all taxonomic groups. I'd suggest picking one level (eg Family or Genus) and using a bar plot (or something similar) to show proportions/reads in the control versus the treated larvae. This would also better represent the information currently in 3B, which is not really readable (the scale bar is also not very discriminating).

Decision letter (RSPB-2021-1819.R0)

07-Sep-2021

Dear Dr Galarza:

Your manuscript has now been peer reviewed and the reviews have been assessed by an Associate Editor. The reviewers' comments (not including confidential comments to the Editor) and the comments from the Associate Editor are included at the end of this email for your reference. As you will see, the reviewer has raised some issues with your manuscript and we would like to invite you to revise your manuscript to address them.

When submitting your revision please upload a file under "Response to Referees" in the "File Upload" section. This should document, point by point, how you have responded to the reviewers' and Editors' comments, and the adjustments you have made to the manuscript. We

require a copy of the manuscript with revisions made since the previous version marked as 'tracked changes' to be included in the 'response to referees' document.

Research ethics:

Use of animals and field studies:

It is a condition of publication that you make available the data and research materials supporting the results in the article (<https://royalsociety.org/journals/authors/author-guidelines/#data>). Datasets should be deposited in an appropriate publicly available repository and details of the associated accession number, link or DOI to the datasets must be included in the Data Accessibility section of the article (<https://royalsociety.org/journals/ethics-policies/data-sharing-mining/>). Reference(s) to datasets should also be included in the reference list of the article with DOIs (where available).

If you wish to submit your data to Dryad (<http://datadryad.org/>) and have not already done so you can submit your data via this link [http://datadryad.org/submit?journalID=RSPB&manu=\(Document not available\)](http://datadryad.org/submit?journalID=RSPB&manu=(Document%20not%20available)), which will take you to your unique entry in the Dryad repository.

Online supplementary material will also carry the title and description provided during submission, so please ensure these are accurate and informative. Note that the Royal Society will not edit or typeset supplementary material and it will be hosted as provided. Please ensure that

the supplementary material includes the paper details (authors, title, journal name, article DOI). Your article DOI will be 10.1098/rspb.[paper ID in form xxxx.xxxx e.g. 10.1098/rspb.2016.0049].

Please submit a copy of your revised paper within three weeks. If we do not hear from you within this time your manuscript will be rejected. If you are unable to meet this deadline please let us know as soon as possible, as we may be able to grant a short extension.

Best wishes,
Professor Hans Heesterbeek
mailto: proceedingsb@royalsociety.org

Associate Editor Board Member

Comments to Author:

Thank you for addressing concerns raised by the reviewers. The manuscript is substantially improved. It has been seen again by one of the initial reviewers, who has raised a few additional questions that I would like you to also address.

Reviewer(s)' Comments to Author:

Referee: 1

Comments to the Author(s).

The new results in this resubmission, especially data on changes to the microbiome during antibiotic treatment, significantly improve the manuscript. They also lead to a few new questions and suggestions.

1) Please clarify that defensive secretions contain microbes that are representative of the gut population. I assume this is the case, but I don't know the system well enough to be sure.

2) Antibiotics can cause changes to total bacterial densities, microbiome composition or both. Although the data focus on microbial diversity, the response letter and Discussion (e.g. Line 348) seem to focus on "bacterial lode" (which I assume means total density/abundance).

a. Abundance isn't measured, as far as I can tell. This should be clarified.

b. Several diversity indices are shown in Fig 2. As these are all concordant, it isn't really necessary to show them all. More importantly, please spell out the relevance of diversity per se (even if this is only a guess). Is the argument that reduced diversity results in the loss of bacterial groups with possible pathogens, or that higher diversity requires a more "expensive" immune response?

c. Fig 3 is not very informative as presented. Both somewhat duplicate the information in Fig 2 by showing that diversity is lower across all taxonomic groups. I'd suggest picking one level (eg Family or Genus) and using a bar plot (or something similar) to show proportions/reads in the control versus the treated larvae. This would also better represent the information currently in 3B, which is not really readable (the scale bar is also not very discriminating).

Author's Response to Decision Letter for (RSPB-2021-1819.R0)

See Appendix B.

RSPB-2021-1819.R1

Review form: Reviewer 1

Recommendation

Accept as is

Scientific importance: Is the manuscript an original and important contribution to its field?

Good

General interest: Is the paper of sufficient general interest?

Good

Quality of the paper: Is the overall quality of the paper suitable?

Good

Is the length of the paper justified?

Yes

Should the paper be seen by a specialist statistical reviewer?

No

Do you have any concerns about statistical analyses in this paper? If so, please specify them explicitly in your report.

No

It is a condition of publication that authors make their supporting data, code and materials available - either as supplementary material or hosted in an external repository. Please rate, if applicable, the supporting data on the following criteria.

Is it accessible?

Yes

Is it clear?

Yes

Is it adequate?

Yes

Do you have any ethical concerns with this paper?

No

Comments to the Author

The authors have addressed my few questions. No further changes are needed.

Decision letter (RSPB-2021-1819.R1)

23-Sep-2021

Dear Dr Galarza

I am pleased to inform you that your manuscript entitled "Antibiotics accelerate growth at the expense of immunity" has been accepted for publication in Proceedings B.

Data Accessibility section

Open Access

Paper charges

Sincerely,

Professor Hans Heesterbeek

Associate Editor:

Board Member: 1

Comments to Author:

(There are no comments.)

Board Member: 2

Comments to Author:

(There are no comments.)

Appendix A

Dear Dr Galarza:

I am writing to inform you that your manuscript RSPB-2021-0111 entitled "Antibiotics accelerate growth at the expense of immunity" has, in its current form, been rejected for publication in Proceedings B.

This action has been taken on the advice of referees, who have recommended that substantial revisions are necessary. With this in mind we would be happy to consider a resubmission, provided the comments of the referees are fully addressed. However please note that this is not a provisional acceptance.

Sincerely,

Professor Hans Heesterbeek
mailto: proceedingsb@royalsociety.org

Author's reply:

Dear Prof. Heesterbeek,

Thank you for the opportunity to resubmit a revised version of our manuscript entitled "Antibiotics accelerate growth at the expense of immunity" to Proceedings B. We are very happy to see that both reviewers and the associate editor find our paper as an important contribution to the field. We are very thankful to the reviewers and associate editor. Their critical review and helpful comments have helped us substantially in improving our paper. We have carefully gone through all suggestions they made and give below a detailed point by point answer to all questions they raised. The most important change we have done is added

the analyses of the effects of antibiotics on the microbiome as requested by referee 1 (pages 6-8). Our results show that antibiotics decrease significantly the bacterial load in the treated moths (lines 540-570). This could help explain the increased growth via reduced pathogen burden. As the bacterial load is reduced, resources could be relocated to growth instead to upkeeping the immune system. This notion is supported by the RNA-seq data. We hope that our revised version now meets the high standards of the journal.

Associate Editor

Comments to Author:

Thank you for submitting your manuscript “Antibiotics accelerate growth at the expense of immunity” to Proceedings B. I have now received two reviews and evaluated the manuscript myself. We all see clear importance in the question investigated, however, less clear is the level of evidence that this study provides and its ability to distinguish between multiple explanations; see in particular comments from reviewer 1 regarding the effects of antibiotics on bacterial infections. In addition to points raised by the reviewers I’m wondering why only needle prick was used for the control group, rather than injection of distilled water, and if this could have influenced the results.

Author’s reply:

*We considered several approaches to obtain a reliable and comparable control group. We decided to prick only and not to inject distilled water into the controls. This is because injecting distilled water could potentially induce changes in osmolarity that could confound the results. Changes in osmolarity of the hemolymph induces activation of hormonal pathways linked to homeostasis maintenance and fluid secretion in the Malpighian tubules. Among the secondary messengers in these pathways includes cAMP (Audsley et al., 1993 *Experimental Biology*, 178, 231-243) and other peptides that have dual roles in immune function and maintenance of homeostasis (Liu et al., 2017, *BMC Genomics*, 18, 974, doi.org/10.1186/s12864-017-4355-5). Their increased presence when filtering hemolymph in response to changes in the osmotic balance (Ruiz-Sanchez et al., 2015, *Journal of Insect Physiology*, 82, 92-98), can lead to altered expression of immune genes and immune functioning (Marin et al., 2005, *Journal of Insect Physiology*, 51 (5), 575-586). Therefore, to avoid overcomplicating the expression of immune genes in the control group of larvae as a result of these processes, we only pierced the exoskeleton with sterile needles to control for exoskeleton injury. Changing the osmolarity of the haemolymph by injecting distilled water into the larvae could have made comparisons of immune gene expression more unreliable, so we tried to avoid this risk.*

Reviewer(s)' Comments to Author: Referee: 1

Comments to the Author(s)

The aim of this paper is to test whether antibiotic treatment can alter the trade-off between growth and immunity in wood tiger moths. This is interesting as it bears on the use of antibiotics as growth promoters in animal husbandry. Using a split brood design, the authors fed half of the larvae in a set of broods with antibiotics and then measured several life-history traits as well as transcriptional profiles. They found that antibiotic fed larvae pupated and reached adulthood a few days earlier than untreated larvae; however, egg number was unchanged and treated adult females were lighter. They also found that several genes associated with immunity were down-regulated in treated larvae while genes associated with growth were upregulated.

The results are sound, but I'm not convinced they really increase understanding of why antibiotics lead to growth promotion. It also isn't known if the link between growth and immunity is causal. First, the issue of growth promotion is undoubtedly tied to the microbes in the moth. Antibiotics can reduce bacterial numbers or change microbiome composition, either of which could affect immunity. Any effects of the antibiotic treatment on bacteria were unfortunately not measured. Second, growth promotion is strongly dependent on antibiotic concentrations, how much is added and for how long the dose remains high enough to be effective. As these weren't manipulated, it isn't clear if the dose was sub-therapeutic, therapeutic or even toxic. Without estimates on the bacteria that tetracycline and ciprofloxacin target, this isn't known. I don't like asking for additional experiments, but in this case, I strongly feel that the story is very incomplete without data on the bacteria.

Author's reply:

We thank the reviewer for the encouraging comments. We fully agree, it is important to show that the antibiotic treatment had a significant influence on the bacterial community. In this revised version we have profiled the bacterial community of moths from this experiment as requested by the reviewer. Briefly, using the 16S ribosomal as gene marker, we observed a significant reduction in bacterial abundance and diversity in moths treated with antibiotics relative to untreated (Figure 2). The reduction was substantial at all taxonomic levels (Figure 2). We have added these new analyses on pages 5-6. Interestingly, some bacteria removed in the antibiotic treatment are known to be pathogenic to several insects (i.e. entomopathogenic). In particular, bacteria from the genera *Haemophilus*, *Shewanella*, *Bacillus*, *Streptococcus*, *Acinetobacter*, and *Corynebacterium* are known to be pathogenic to several *Lepidoptera*. All these genera were either removed or depleted in the antibiotic treatment (Figure 3). We discuss these results in lines 440-462 and provide a list of bacterial taxa known to be pathogenic to *Lepidoptera* in supplementary table 6.

Figure 3. A) Number of bacterial taxa per taxonomic level not found in the antibiotic treatment B). Abundance and diversity of bacteria per taxonomic level. Bacterial taxa not found in the antibiotic treatment are highlighted in grey. A high-resolution image can be found in supplementary figure 3.

*Regarding the dose administered; usually (and wrongly) only successful methods are reported. We performed several previous attempts to determine a non-lethal antibiotic dose for our experiment. In our first attempts, antibiotics were administered orally by dipping food leaves (dandelion, *Taraxacum* ssp.) into the antibiotic solution at different concentrations (1mg/ml, 0.5 mg/ml, 0.01mg/ml) following the methods of Zha et al. 2014 (*Genetics and Molecular Biology*, 37(3), 573-580). In all cases, mortality was very high with only <3% of larvae reaching adulthood. Hence, we opted for a more controlled intermediate dose (0.04 mg/ml) administered via injection. We have now clarified this in lines 91-95.*

A few more detailed questions are given below:

1) A key issue is the assumption that changes to a particular set of genes are associated with a change in a phenotype of interest. How strong is the assumption that changes to a collection of immunity genes has actually had an impact on immunity?

Author's reply:

*This is a fair point and certainly requires more future work. In humans, it has been shown that changes in immunity that result from antibiotic treatment can lead to increased susceptibility to infection. Many changes in immunity have been observed at the molecular level in response to antibiotics, including reduced secretion of antimicrobial peptides and changes in T helper cell populations (Willing, B., Russell, S. & Finlay, B. *Nat Rev Microbiol* 9, 233–243, 2011). Evidence from other vertebrate model systems (mice) shows that antibiotics can inhibit respiratory activity in immune cells and consequently impair their phagocytic activity (Yang et al., 2017: *Cell & host environment* VOLUME 22, ISSUE 6, P757-765.E3). In insect model species (silkworm), antibiotics can cause various physiological (i.e. antioxidant enzyme activities) and histological damage (Li, et. al. *Chemosphere*, 248, 126019, 2020). In previous research from our own group, we have shown that antibiotics can induce significant changes in key life-histories (including immunity) in the wood tiger moth (Dickel et al., *Journal of Insect Science*, 16(1), 46, 2016). Hence, in general, it can be safely assumed that the impact of antibiotics in immune genes can be reflected in immunity.*

2) Related to the above, gene expression data are relative, i.e. treated to untreated. Increased relative expression can therefore be caused by an absolute increase in the treatment group or an absolute decrease in the untreated group (and vice versa). As immunity genes, and obviously growth related genes, change naturally during development, it would be valuable to comment on/show absolute as well as relative differences through time.

Author's reply:

Yes, the reviewer is right, gene expression changes naturally with development. In supplementary figure 4 we now show the absolute gene expression (i.e. not only differentially expressed genes) throughout larval development in both treatments. We considered as expressed genes only those that had >10 counts in each sample. The absolute number of

expressed genes increased with development in both treatments. More genes were expressed late in development (i.e. injection 3) compared to intermediate stages of development (i.e. injection 2) in both treatments. Interestingly, more genes were expressed early in development (i.e. injection 1) in the antibiotic treatment compared to the controls. This is probably due to a primal reaction to the antibiotics. However, the absolute expression in later stages followed the same trend in both treatments. Hence, it is unlikely that the differences observed in relative gene expression could be caused by an increase/decrease in absolute gene expression in one of the treatments. We introduce this comment in lines 219-225 and added the supplementary figure 4.

Supplementary figure 4. Absolute gene expression in wood tiger moth (*Arctia plantaginis*) larvae after administration of 1-3 antibiotic injections.

3) Adult females in the antibiotic group are actually lighter, in contrast to the expectation that these agents are growth promoters. Is it possible that they are lighter simply because their developmental time is shorter, whereas the untreated slower larvae have a few extra days to feed? Given this, it might be worth checking if time to pupation and female weight are correlated, and if this scaling varies between the treatment groups (especially given the fairly sizable variance in both variables). The same question applies to the scaling between mass and egg number.

Author's reply:

Indeed, as often is the case in insects, developmental time and adult size/mass correlate. We have assessed for possible correlations and found a weak but significant correlation between time to pupation and female weight in the antibiotic treatment, as the reviewer rightfully suspected (Supplementary figure 5). However, such correlation was not observed in females from the control treatment. The same patterns were observed when assessing the relationship between the number of eggs and female weight which showed a strong correlation in the antibiotic treatment only (Supplementary figure 5). Thus, it can be suggested that females in the antibiotic treatment could be lighter simply because a shorter development period. This is now discussed in lines 309-322. However, this interpretation should be taken with caution since the control females were also 'treated' by pricking them. In a recent study in this species, we simulated non-lethal predation attacks by pecking larvae with tweezers throughout their development and measured different life-histories (Almeida et al., 2021. Predator-Induced Plasticity on Warning Signal and Larval Life-History Traits of the Aposematic Wood Tiger Moth, *Arctia plantaginis*. *Frontiers in Ecology and Evolution*, 9, 412). We found that the pecking treatment did not have an effect on the larval development time.

Supplementary figure 5. Weight of wood tiger moth (*Arctia plantaginis*) adult females and its correlation (Spearman) to their time spent as pupae and their number of eggs produced.

4) Mortality data are entirely overlapping until the last instar. Is there any reason why 35% of the larvae in the untreated group suddenly die? Greater survival could be driven by the direct effects of the antibiotics in the antibiotic-treated larvae. But it's striking that this increased mortality occurs in the group with the higher expression of immune genes. This could be further explained.

Author's reply:

In laboratory conditions larval mortality is typically 10-20% with the highest mortality peaks happening early and late in development. The mortality in our experiment was ~10% higher most likely due to pricking itself. This is now clarified in lines 320-322.

Referee: 2

Comments to the Author(s)

This paper investigates whether antibiotic-induced growth promotion results from a trade-off between immune investment and growth. The authors use the wood tiger moth (*Arctia plantaginis*) to test this hypothesis. Larvae treated with broad-spectrum antibiotics showed a faster growth rate, up-regulation of growth genes, and down-regulation of genes involved in immunity. Overall, this is an interesting paper that investigates a question important to agriculture as well as to life history evolution.

Author's reply:

Thank you!

Major comments:

1. The authors observe that antibiotic treatment leads to down-regulation of immune genes and increased growth. It seems to me that there are at least two possible explanations for this, which are not mutually exclusive: 1) decreased caloric investment in immunity allows for increased investment in growth (life history trade-off), or 2) decreased nutrient uptake by bacteria frees up energy for increased growth (reduced pathogen burden). The authors focus on the first possibility, but it would be worthwhile to add discussion of alternative explanations.

Author's reply:

The reviewer is absolutely right. In this revised version we include additional analyses of bacterial community profiles from moths of both treatments. Using the ribosomal 16S as marker gene we observe a significant reduction in bacterial load in moths treated with antibiotics, which is in line with the reviewer's pathogen burden hypothesis. We introduced these new analyses in pages 5-6, and elaborate on their possible role in the observed increased growth in lines 440-462. Moreover, we provide a list of bacterial taxa known to be pathogenic to Lepidoptera in supplementary table 6.

Minor comments:

1. For Figures 1&2, please add a brief description of what the box plots show in the legend (e.g. median and quartiles shown for each group).

Author's reply:

Figures 1&2 are now 4&5 in this revised version. We have added the description in the legend as requested.

2. Figure 4: Please increase the font size for the labels in this figure.

Author's reply:

Figure 4 is now Figure 1 in this revised version. We have increased the font as requested

3. It would be helpful to briefly discuss whether the observed changes in growth and gene expression are mediated by the removal of bacteria or direct physiological effects of antibiotics on invertebrate cells. For example, is growth promotion observed when insects are raised in germ-free conditions? This would suggest that decreased bacterial load is important to the mechanism of growth promotion. In vertebrates, antibiotics like macrolides can also have direct physiological effects on cells in the absence of bacteria.

Author's reply:

The reviewer makes a very good point. Overall, evidence from germ-free and gnotobiotic-reared insects points towards a negative impact on growth in the absence-depletion of associated microbes. For instance, in bacteria-free mosquitoes larvae take longer to develop and produce smaller adults than bacteria-colonised larvae (Correa et al., 2018, Nature Communications, 9(1), 4464). The same observation of larval depressed development and reduced body mass has been observed in Coleoptera Habieza et al., 2019, Frontiers in Microbiology, 10, 1212), Diptera (Hassan et al., 2020, Frontiers in Microbiology, 11, 2760) and also in Lepidoptera species (Somerville, et al., 2019, Insects, 10, 89). In this revised version we show that the removal of bacteria does impact immune processes and in consequence, growth. Thus, living microorganisms appear to strongly influence insect larval development, as the reviewer correctly suggests, but pinpointing the exact mechanisms is a complex task. We introduce these new analyses in pages 5-6 and discuss the results in lines 440-462.

4. Data availability is listed for sequencing data but not life-history data (e.g. growth data and egg counts). Please make these data available.

Authors' reply:

We apologize for this omission. The data is now available in Supplementary table 7.

5. Linear mixed effects models are used here to analyze count data (e.g. egg numbers). Count data often violate the assumptions of a linear model, and a Poisson model might be more appropriate. Have model diagnostics shown that the linear model is appropriate to these data? If not, I would suggest a Poisson distribution or similar alternative.

Authors' reply:

The reviewer is absolutely correct, count data often violates assumptions of linear models (i.e. linearity, normality of residuals, homoscedasticity) and violations of these assumptions should be tested. We apologize for this omission and have now performed several tests including normality test and diagnostic plots to evaluate our model's suitability. Shapiro-Wilk normality tests indicated that the distribution of our egg-number data is not significantly different from a normal distribution ($W = 0.968$, $p\text{-value} = 0.598$). This was in agreement with the Normal Q-Q plot in which the residuals appeared normally distributed (Supplementary figure 6). Linearity and homogeneity of variance of the residuals (homoscedasticity) were confirmed by inspecting the residuals vs fitted and scale-location plots respectively. Three possible influential values (i.e. high leverage values) were identified by examining the Residuals vs Leverage plot. These were removed from the dataset and we repeated the linear model. The results changed if absolute values but not in significance. Previously: estimate 0.02327, $P = 0.2903$, and with high-leverage values removed: estimate 0.03308, $P = 0.2211$. We have now introduced into the main text the process of model validation in lines 191-195, and the new significance values in lines 315-317. We thank the reviewer for pointing this to us.

Supplementary figure 6. Diagnostic plots assessing for departures of linear model assumptions.

Appendix B

Dear Prof. Heesterbeek,

Thank you for the opportunity to revise our manuscript entitled "*Antibiotics accelerate growth at the expense of immunity*" (RSPB-2021-1819).

We are grateful to both the associate editor and the reviewers who agreed to re-asses our revised manuscript. Their comments and suggestions significantly enhanced the manuscript. Below we address the few comments made by Reviewer 1 to our revised manuscript. We hope that our manuscript now meets the criteria for publication in Proceedings of the Royal Society B.

On behalf of all co-authors,

Dr. Juan A. Galarza
Dept. of Biological and Environmental Sciences
University of Jyväskylä, Finland

Reviewer 1:

1) Please clarify that defensive secretions contain microbes that are representative of the gut population. I assume this is the case, but I don't know the system well enough to be sure.

Author's response:

We thank the reviewer for this comment. We chose not to sample the gut from these adults because of the temporal and transient nature of gut microbes in Lepidoptera (see below). It is unclear what the effect of antibiotics could have on such an unstable microbiota. Moreover, wood tiger moth adults don't feed and have only a vestigial gut-like structure remnant from the larval stage. Also, we did not want to remove the gut from larvae because we needed to grow them to adults. Most importantly, the microbial data we present is to show that antibiotics worked and removed a significant number of bacteria. The point of our study was to show the genetic trade-off between growth and immunity that may be caused by the direct effect of altered microbiome or removal of potentially harmful bacteria.

While it is true that studies over the past decade have revealed that gut bacteria can have multiple impacts on host's fitness; as technology improves in quantifying and analysing microbial communities, it is being revealed that gut bacteria are not as ubiquitous and essential as previously thought. In particular, recent studies in Lepidoptera show evidence that gut bacteria can be sparse, transient, or unpredictable (i.e. Hammer et al., 2017. Caterpillars lack a resident gut microbiome. Proceedings of the National Academy of Sciences, 114(36), 9641; Paniagua et al., 2018. Bacterial Symbionts in Lepidoptera: Their Diversity, Transmission, and Impact on the Host. Frontiers in Microbiology, 9, 556). Hence, we sampled the microbes from the moth's defensive secretion, a type of reflex-bleed mechanism from the hemolymph, which is a key survival trait for the species (Rojas et al., 2017. How to fight multiple enemies: target-specific chemical defences in an aposematic moth. Proceedings of the Royal Society B: Biological Sciences, 284) and thus, presumably conserved, including its associated microbiota. We have clarified this in lines 148-153.

Reviewer 1:

2) Antibiotics can cause changes to total bacterial densities, microbiome composition or both. Although the data focus on microbial diversity, the response letter and Discussion (e.g. Line 348) seem to focus on "bacterial load" (which I assume means total density/abundance).

Author's response:

The reviewer is correct, the term needs clarification since it can have different meanings depending on the field. We have now clarified in line 353 that by -bacterial load- we refer to the type (i.e. diversity) and amount (i.e. abundance) of bacteria taxa.

a. Abundance isn't measured, as far as I can tell. This should be clarified.

Author's response:

Abundance is measured as number of reads per taxa. This is now better stated in lines 168-170. Figure 3B shows the number of reads/family, and supplementary figure 3 shows the

abundance of all taxonomic levels in the scale. The width of the scale is proportional to the number of reads.

b. Several diversity indices are shown in Fig 2. As these are all concordant, it isn't really necessary to show them all. More importantly, please spell out the relevance of diversity per se (even if this is only a guess). Is the argument that reduced diversity results in the loss of bacterial groups with possible pathogens, or that higher diversity requires a more "expensive" immune response?

Author's response:

The figure has been replaced with the one below showing only the two diversity indices following the reviewer's request. We show two diversity indices because each tends to approach quantifying diversity in a different manner, with one index having its weakness covered by another index and vice versa. For instance, Chao1 index provides diversity estimates including estimations for unobserved taxa, whereas Shannon's index considers the relative abundances of taxa and community evenness in its calculations. We now clarify this in lines 162-169.

Figure 2. Bacterial alpha diversity indices associated to wood tiger moth (*Arctia plantaginis*) treated with antibiotics and untreated controls.

As for the relevance of diversity -per se-, the reviewer is right on his/her rationale. Both arguments are correct and not mutually exclusive. A reduced bacterial diversity in the antibiotic group does result in the loss of pathogenic taxa, as indicated in lines 445-447. In parallel, an increased bacterial diversity in the control group may require higher immune responses to constrict abundances of multiple species that could upset homeostasis if left to multiply unchecked. These taxa would not have to be pathogens (even opportunistic ones), merely it could upset chemical balances in the microbiome and subsequent potential functions it provides to the moth. We have now included this notion in lines 450-455.

c. Fig 3 is not very informative as presented. Both somewhat duplicate the information in Fig 2 by showing that diversity is lower across all taxonomic groups. I'd suggest picking one level (eg Family or Genus) and using a bar plot (or something similar) to show proportions/reads in the control versus the treated larvae. This would also better represent the information currently in 3B, which is not really readable (the scale bar is also not very discriminating).

Author's response:

We apologize for the unclarity of this figure. We have produced a new figure according to the reviewer request (below), and Figure 3B is now presented as Supplementary figure 3.